# Delay in recovery of the Antarctic ozone hole from unexpected CFC-11 emissions

S.S. Dhomse [1,2], W. Feng [1,3], S.A. Montzka [4], R. Hossaini [5], J. Keeble [6,7], J.A. Pyle [6,7], J.S. Daniel[4] & M.P. Chipperfield [1,2]*

The Antarctic ozone hole is decreasing in size but this recovery will be affected by atmospheric variability and any unexpected changes in chlorinated source gas emissions. Here, using model simulations, we show that the ozone hole will largely cease to occur by 2065 given compliance with the Montreal Protocol. If the unusual meteorology of 2002 is repeated, an ozone-hole-free-year could occur as soon as the early 2020s by some metrics. The recently discovered increase in CFC-11 emissions of ~ 13 Gg yr$^{-1}$ may delay recovery. So far the impact on ozone is small, but if these emissions indicate production for foam use much more CFC-11 may be leaked in the future. Assuming such production over 10 years, disappearance of the ozone hole will be delayed by a few years, although there are significant uncertainties. Continued, substantial future CFC-11 emissions of 67 Gg yr$^{-1}$ would delay Antarctic ozone recovery by well over a decade.

[1] School of Earth and Environment, University of Leeds, Leeds LS2 9JT, UK. [2] National Centre for Earth Observation (NCEO), University of Leeds, Leeds LS2 9JT, UK. [3] National Centre for Atmospheric Science (NCAS), University of Leeds, Leeds LS2 9JT, UK. [4] Earth System Research Laboratory, Global Monitoring Division, National Oceanic and Atmospheric Administration (NOAA), Boulder, USA. [5] Lancaster Environment Centre, Lancaster University, Lancaster, UK. [6] Department of Chemistry, University of Cambridge, Cambridge CB2 1EW, UK. [7] National Centre for Atmospheric Science (NCAS), University of Cambridge, Cambridge CB2 1EW, UK. *email: M.Chipperfield@leeds.ac.uk

Depletion of the stratospheric ozone layer by chlorine and bromine species has been a major environmental issue since the early 1970s[1,2]. Following controls on the production of the long-lived halocarbons that transport chlorine and bromine to the stratosphere, atmospheric concentrations of most of them are now decreasing[3], and the ozone layer is expected to recover over the course of this century[4]. Decreases in the stratospheric loading of chlorine and bromine have been observed[5,6], and there are signs of a consequent increase in ozone in the upper stratosphere and the total column[7,8].

The most significant signal of anthropogenic ozone depletion occurs in the Antarctic in spring—the so-called Antarctic ozone hole. The size of the hole is typically quantified using a range of metrics including minimum column ozone, area contained within a particular value of column ozone and ozone mass deficit[4]. From these metrics, it is clear that the hole has also stopped increasing in size and there are signs of recovery[9–12]. However, even with full compliance with the Montreal Protocol, it is expected that the Antarctic ozone hole will persist for many decades into the future.

The most recent comprehensive assessment of ozone return dates was performed by Dhomse et al.[13], using results from the chemistry–climate modelling initiative (CCMI) as input to the 2018 WMO Assessment[4]. They used results from 20 coupled chemistry–climate models (CCMs) to make a best estimate of the dates at which future ozone levels would return to their 1980 values for polar, mid-latitude and tropical regions. Based on this they reported that October column ozone in Antarctic (60ºS–90ºS) would return to 1980 values in 2060 (with a 1σ uncertainty of 2055–2066) and for March in the Arctic (60ºN–90ºN) by 2034 (2025–2043). The much earlier return date in the Arctic is due to the smaller depletion and large variability, in conjunction with climate change, which was estimated to have only a small (2-year) effect on the much larger Antarctic loss. This metric of return to a 1980 value is a straightforward concept but it is difficult to estimate and needs to be interpreted with caution. In order to derive values from CCMs, Dhomse et al. (and similar earlier studies[14]) employed significant smoothing and averaging of the modelled ozone. As a result, the influence of interannual dynamical variability is not included in the uncertainty ranges given above. In addition, in regions where ozone values return asymptotically to the 1980 reference value, a small change in ozone can lead to a large change in return date. Indeed, until and unless ozone values return to the 1980 values then recovery could be deemed not to have happened.

In addition to uncertainties in how to quantify recovery of the ozone layer, a number of factors pose a threat to its expected timescale. Montzka et al.[15] recently reported that since the mid-2000s, atmospheric CFC-11 has not been declining as expected, and this discrepancy became particularly striking after 2010, the year production of CFC-11 was reportedly phased out. Their results suggest new emissions[16] that linked to unreported production have occurred in recent years. Based on current atmospheric abundances, CFC-11 still contributes about one-quarter[15] of anthropogenic chlorine (about one-fifth of all chlorine[3]) reaching the stratosphere, so these results could have significant implications for recovery of the ozone layer. Moreover, because nearly all produced CFC-11 eventually escapes to the atmosphere, the impact of this apparent renewed use of CFC-11 on stratospheric ozone ultimately depends on the total amount of new, unreported production, which is currently unknown. If the detected unexpected emissions arise from CFC-11 produced for an emissive use, no large increases in CFC-11 production or banks would be implied, and one would expect the emissions to diminish rapidly if use were terminated. However, because past CFC-11 use was primarily for blowing closed-cell insulating foam that retained most of the CFC[17], the detected magnitude of the new, unexpected emissions could imply much larger CFC-11 production quantities and therefore large future emissions.

Recent studies have also suggested that increasing atmospheric emissions of very short-lived substance species (VSLS), which are not controlled by the Montreal Protocol, may cause a delay to ozone layer recovery. Hossaini et al.[18] reported increasing emissions of dichloromethane ($CH_2Cl_2$) based on atmospheric observations from the NOAA global surface network. Using a CCM to investigate different assumptions of the future evolution of $CH_2Cl_2$, they estimated a delay in the return of Antarctic ozone to 1980 levels of 5 years for constant future $CH_2Cl_2$ concentrations, compared with zero atmospheric $CH_2Cl_2$, and up to 30 years for an extreme sensitivity study of constantly increasing $CH_2Cl_2$ concentrations. However, those timescales for ozone recovery need to be interpreted with caution as the likelihood for these scenarios to be realised is unknown and the timescales depend on the slow convergence of ozone to a reference recovery baseline. Recently, Fang et al.[19] reported increases in the atmospheric abundance of the VSLS chloroform ($CHCl_3$). They also used the model results of Hossaini et al. to estimate the impact of sustained $CHCl_3$ growth on ozone recovery and thereby derived significant delays in 1980 return dates, but these values will have the same caveat of a small change in ozone, causing a large difference in return date, as discussed above. These results need to be reassessed for the impact of realistic amounts of chlorinated VSLS in the context of other changes to chlorine source gases and for a wider range of ozone-hole recovery metrics.

Therefore, although ozone recovery is underway, there is uncertainty in how it will progress in the future and how it should be reported. The date for the atmosphere to return to a specified state does not take account of variability in that pathway or on the impact of other transient factors before the final return date. For the Antarctic ozone hole in particular, there are other measures of its size, which may give a different perspective on recovery.

In this paper, we use a detailed atmospheric chemical transport model (CTM) to investigate the impact of meteorological variability and non-compliant or uncontrolled chlorine source gas emissions on polar ozone recovery. We quantify the persistence of the Antarctic ozone hole by a range of different metrics, which we show need to be interpreted carefully. We are interested in how long these metrics suggest an ozone hole will occur and how soon, given meteorological variability, we may experience a year without a hole. We also investigate how much longer the Arctic may be susceptible to large ozone loss based on recent extreme meteorology observed in the year 2011 (with extensive occurrence of low temperatures that are conducive to ozone loss). For both cases, we explore the impact on this recovery of the unexpected increase in emissions of CFC-11, under different scenario assumptions. We compare this with the impact of uncontrolled chlorinated VSLS that also enhance polar ozone loss through the increase in stratospheric chlorine. Given the large uncertainties in the CFC-11 emissions related to their source, their main application and future trends, our aim is to test example scenarios and quantify how the impact on ozone varies with the timing and magnitude of the emissions. Our 3D model results can therefore be scaled to assess the impact on column ozone of other total CFC-11 emissions.

## Results

**Chlorine scenarios.** Figures 1 and 2 show estimates of CFC-11 emissions and the corresponding mean global atmospheric mixing ratios from a box model (see the 'Methods' section). The 2018 WMO Assessment[20] baseline mixing ratio scenario uses a combined atmospheric observation record up to the beginning of

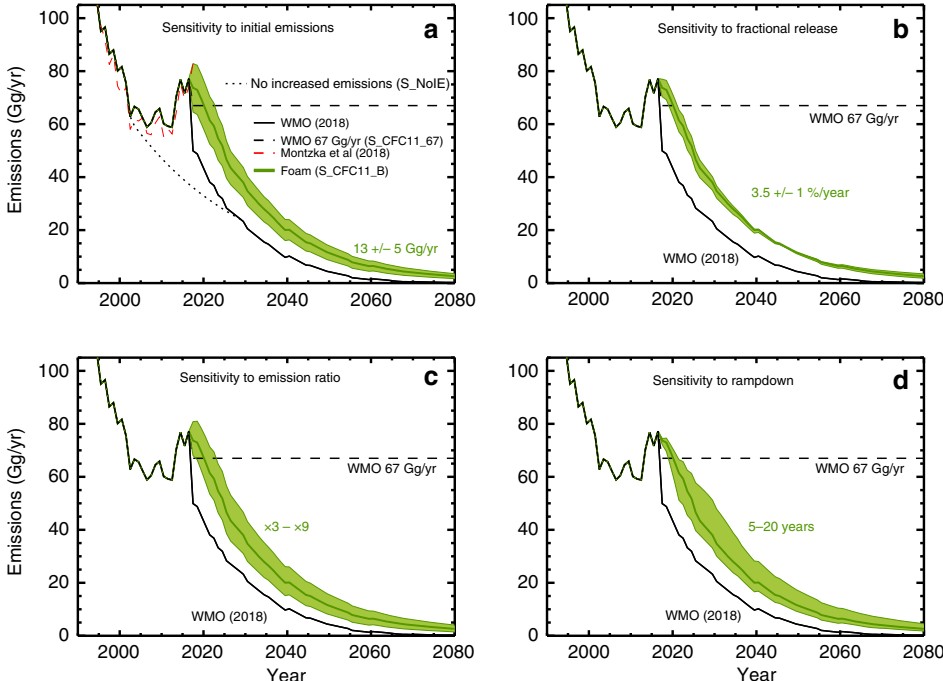

**Fig. 1 Past and potential future emissions of CFC-11.** Estimated emissions of CFC-11 for the past derived from atmospheric measurements and for future scenarios with different assumptions. **a** Five scenarios including the World Meteorological Organisation (WMO) (2018)[20] baseline scenario (solid black line) and an assumption of constant 67 Gg yr⁻¹ emissions (dashed line, S_CFC11_67). Also shown is a scenario with decreasing emissions from an estimate of the 2002 bank (dotted line, S_NoIE). Emissions based on box model simulations for unreported CFC-11 production for foam use are shown in green (solid line, S_CFC11_B). The green shading indicates the sensitivity range for S_CFC11_B for initial emissions ranging from 8 to 18 Gg/yr (see Methods for assumptions). **b** Similar to panel **a**, for WMO (2018) baseline and S_CFC11_67 scenarios with green shading showing sensitivity of scenario S_CFC11_B for fractional release ranging from 2.5 to 4.5%/year. **c** Similar to panel **b**, and showing sensitivity to emission ratios from ×3 to ×9. **d** Similar to panel **b**, and showing sensitivity to rampdown of new production between 5 and 20 years.

2017 and then future projections. We use this mixing ratio scenario to infer emissions for past years, and obtain good agreement with other estimates[15]. Considering future mixing ratios, the simple WMO scenario with constant future emissions of CFC-11 of 67 Gg yr⁻¹ (the average calculated top–down emissions over 2002–2016, scenario S_CFC11_67) produces a much slower decrease in CFC-11 than the baseline scenario, with the global mixing ratio dropping only to 170 ppt in 2080. We have also constructed an additional CFC-11 emission scenario for the recent past based on a constant release fraction from the bank since 2002[15], which is typically assumed in the creation of future scenarios[20] (S_NoIE; no increased emissions). This scenario indicates the path that would have been expected without these post-2010 emissions from unreported production and also earlier emission changes from 2002, which caused a stabilisation of emissions (Fig. 1a). Therefore, it represents an upper limit of the impact of the recent emission changes. Note that with the box model result, we implicitly assume that the recent variations in the CFC-11 decay rate are all due to emissions. Montzka et al.[15] noted that some of the observed variations could be due to influences of atmospheric dynamics, which would imply a smaller emission increase than we have assumed.

To construct alternative future scenarios, we begin with the estimate of new emissions due to unreported production of 13 Gg yr⁻¹ (based on ref. [15]). We relate this to production by using an estimate of the ratio between rapid initial emission and accumulation in the bank. We then assume a yearly fractional release (leakage) rate for this bank. Finally, we assume a timescale for future rampdown of the unreported production (see Methods). For these 4 parameters, we perform sensitivity tests with the box model to investigate the impact on CFC-11 (Figs. 1

and 2). The accumulated CFC-11 emissions scale directly with the key parameters of initial emissions, emission ratio and timescale for rampdown (Supplementary Fig. 1). For the ranges assumed for these values the impact on the return dates is around ±1 year (see Supplementary Results 2). The results are relatively insensitive to the future fractional release as the CFC-11 is eventually emitted to the atmosphere in any case. Given the ongoing international concern about this issue, we anticipate that the unreported production (and associated initial emissions) will likely stop in the near future, so the main uncertainty for ozone recovery would be the cumulative magnitude of post-2010 unreported production and how it was used (i.e. the emission ratio). The scenario (S_CFC11_B) that assumes production for the non-emissive (foam) use that is phased out over 10 years maintains the peak in emissions of 80 Gg yr⁻¹ for a short period and an additional 15 ppt of CFC-11 (45-ppt Cl) in mid–late century (Fig. 2).

**Antarctic ozone hole.** Different metrics are used to assess the size of the Antarctic ozone hole from observations and model experiments, including September and October mean column ozone from 60ºS to 90ºS; ozone-hole area; ozone mass deficit; minimum total column ozone[4]. For all metrics, the observations show the increasing size of the hole from the time of the first data point shown in 1980 until the early 2000s (Fig. 3). Subsequently the metrics show a peak in the size of the hole followed by suggestions of a decrease from around the mid-2000s. Figure 3 also shows the results from a range of model simulations (Table 1). The control model run CNTL, with time-dependent meteorology, agrees well with the observations for the various metrics, showing

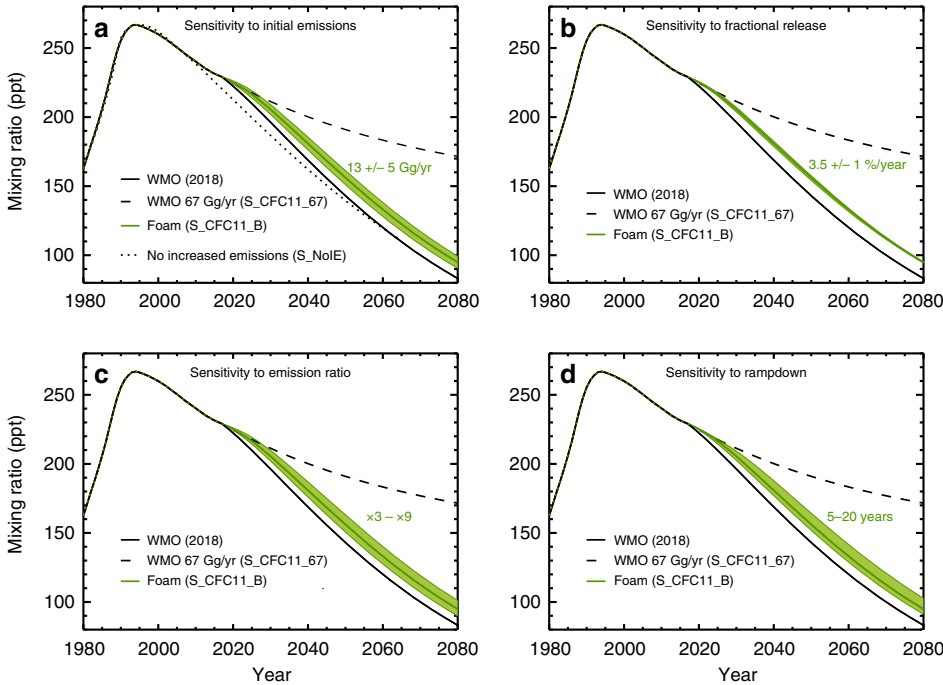

**Fig. 2 Past and potential future concentrations of CFC-11.** As Fig. 1 but for CFC-11 volume-mixing ratio (ppt). **a** Four scenarios including the World Meteorological Organisation (WMO) (2018)[20] baseline scenario (solid black line) and an assumption of constant 67 Gg yr$^{-1}$ emissions (dashed line, S_CFC11_67). Also shown is a scenario with decreasing emissions from an estimate of the 2002 bank (dotted line, S_NoIE). Emissions based on box model simulations for unreported CFC-11 production for foam use are shown in green (solid line, S_CFC11_B). The green shading indicates the sensitivity range for S_CFC11_B for initial emissions ranging from 8 to 18 Gg/yr (see Methods for assumptions). **b** Similar to panel **a**, for WMO (2018) baseline and S_CFC11_67 scenarios with green shading showing sensitivity of scenario S_CFC11_B for fractional release ranging from 2.5 to 4.5%/year. **c** Similar to panel **b**, and showing sensitivity to emission ratios from ×3 to ×9. **d** Similar to panel **b**, and showing sensitivity to rampdown of new production between 5 and 20 years.

that the CTM gives a good quantitative simulation of polar ozone loss, which is driven by chlorine and bromine chemistry. Run R2000 is similar to run CNTL but uses repeating 2000 meteorology and can be extended into the future to investigate the impact of decreasing halogens. This run shows the characteristic signal of ozone recovery with the dates for the different metrics to return to 1980 values ranging from 2063 to 2067 (Table 2). Run fODS, with fixed surface mixing ratios of ozone-depleting substances (ODS), provides a baseline for ozone changes due solely to changes in $N_2O$, $CH_4$ and, in the past, dynamics and aerosol (see Methods). In the future, run fODS also uses repeating 2000 meteorology. The net chemical effect of increasing $N_2O$ and $CH_4$ causes a gradual decrease in the mean September/October column ozone values, which, in itself, extends the return dates of these metrics compared with the direct effect of halogen decreases[21,22].

Results from runs R2002, R2009 and R2010 illustrate the impact of different meteorology on ozone recovery. In reality, interannual variability will change the meteorology from year to year and give rise to a variable signal in, for example, column ozone, seen in runs CNTL and fODS from 1980 to 2016. The background pink line in Fig. 3 from 2018 onwards shows results from the extension of run CNTL with 20-year repeating meteorology. In September, the meteorology for 2002 stands out as extreme (Fig. 3a), while October shows a wider range of meteorology (Fig. 3b). Keeble et al.[23] used a 7-member ensemble of CCM integrations to investigate the range of meteorologically driven annually averaged Antarctic recovery dates. They found the earliest recovery of annual mean ozone to values above those of 1980 at around 2040 and final recovery (after which date ozone

values were always above the 1980 value) in 2060, with an ensemble spread as large as about 15 years. Further analysis of the CCM data gives final recovery dates for October monthly mean column ozone values, averaged from 90°S to 60°S, in the late 2070s, with an ensemble spread ~20 years. This range in return dates is similar to the CTM behaviour for October when run with different meteorologies (Fig. 3b). CTMs that repeat the meteorology of a particular year can complement CCMs by giving a clear signal of the impact of the different meteorologies on the ozone hole as chlorine and bromine levels decline. The disturbed meteorology of run R2002 clearly leads to the smallest ozone hole by all metrics and the earliest returns to 1980 values: this is as early as 2021 for September column ozone, 2031 for October mean column ozone, 2033 for the ozone mass deficit, 2035 for minimum column ozone and 2041 for the ozone-hole area (Table 2). Clearly different metrics can give rise to different return dates due to the different timing of low temperatures (Supplementary Figure 2), and the timing of vortex split and reformation during that winter[24,25]. Runs R2009 and R2010 give ozone return dates that range from 2060–2066 to 2052–2085, respectively. The range of values is much larger for the 2010 meteorology due to the shift of the ozone-hole timing to later in the spring, causing smaller loss in September and larger ozone loss in October (see Supplementary Fig. 2). These results show that using the October return date in assessment studies[13] is not such a clear measure of recovery due to the large interannual variations in vortex conditions during that month[11]. Furthermore, the free-running climate models may capture this variability to different extents, increasing the uncertainty of the multi-model mean return date.

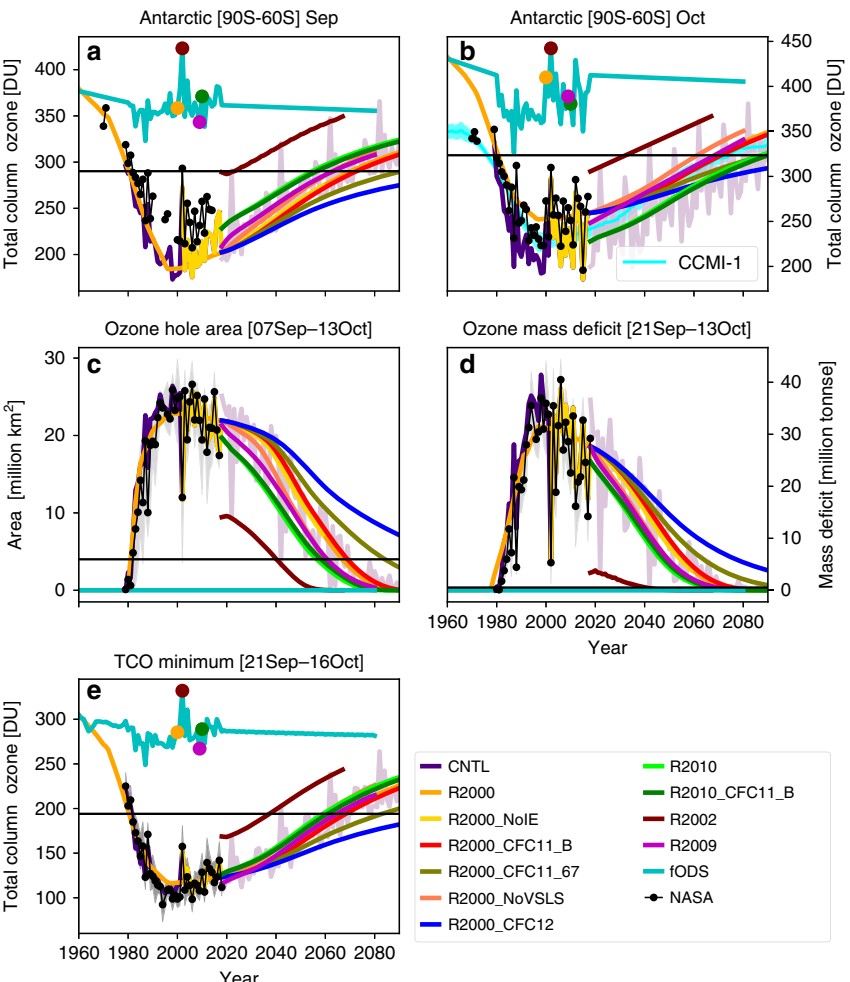

**Fig. 3 Antarctic ozone and metrics quantifying ozone loss as a function of meteorology and additional CFC-11 emissions.** Mean column ozone (DU) averaged from 90°S to 60°S for **a** September and **b** October from TOMCAT simulations CNTL (control), fODS (fixed ozone-depleting substances), R2000, R2002, R2009, R2010 (repeating meteorology from 2000, 2002/2003, 2009/2010 and 2010/2011, respectively), R2000_NoIE (no increased CFC-11 emissions), R2000_NoVSLS (no chlorinated very short-lived substances), R2000_CFC11_67 (with constant CFC-11 emissions of 67 Gg yr$^{-1}$), R2000_CFC11_B and R2010_CFC11_B (with additional CFC-11 emissions from box model for 2000 and 2010/2011 meteorology, respectively) (see legend) from 1960 to 2090. Panel **b** also shows mean (±1σ cyan shading) chemistry-climate modelling initiative (CCMI) results[13]. Estimates of the size of the Antarctic ozone hole using **c** area contained within the 220 DU contour (×10$^6$ km$^2$) (averaged September 7–October 13), **d** ozone mass deficit (×10$^6$ tonnes) (averaged September 21–October 13) and **e** minimum column ozone (between September 21 and October 16). All panels also show observations (black line) from NASA Solar Backscatter Ultraviolet (SBUV) instrument (**a**, **b**) or https://ozonewatch.gsfc.nasa.gov/statistics/annual_data.htm (**c–e**). The coloured dots on the fODS line (panels **a**, **b**) show the years used for simulations R2000, R2002, R2009 and R2010. The pink line in the background in all panels from 2018 to 2090 shows the results of the continuation of run CNTL with 20-year repeating meteorology.

**Arctic ozone depletion.** Springtime ozone depletion in the Arctic is smaller with larger interannual variability compared with the Antarctic[26,27]. Due to this large interannual dynamical variability, it is difficult to determine robust trends in Arctic ozone depletion and recovery, and unlike the Antarctic, there are few relevant metrics with which to assess this. Of interest in the Arctic is predicting how long the region may be susceptible to large chemical ozone depletion as chlorine and bromine levels decline. CCMs predict[13] that mean March Arctic ozone levels will return to 1980 values around 2034 (Fig. 4b), due to the large impact of dynamics, although they may not capture years of large ozone depletion under extreme Arctic meteorology. In recent decades, the year with the conditions most conducive to large chemical ozone loss was winter 2010/2011, when very low temperatures gave rise to large ozone depletion[28]. Figure 4 shows the mean Arctic column ozone in February and March for CTM simulations with a range of meteorology and chlorine scenarios. Simulation R2000 shows that Arctic chemical ozone loss follows

the time variation of chlorine and bromine with a return to 1980 values only around 2080, i.e. much later than the CCMs. For 2010/2011 meteorology, the model predicts only a small increase in mean March column ozone from around 350 DU in March 2010 to around 400 DU by 2100. This is still considerably below the March 1980 baseline of around 450 DU. Hence, whenever years with extremely cold stratospheric conditions occur in this century, the Arctic would be susceptible to ozone depletion, driven by both dynamics and chemical loss[29].

**CFC-11 emissions from unreported production.** Figure 3 also includes the results of simulations that consider different CFC-11 scenarios. The results of run R2000_NoIE, which includes neither the impact of additional inferred post-2010 emissions[15,16], nor the impact of stabilisation of emissions from 2002 to 2010, are very similar to run R2000 (lines essentially overlapping). This shows that the impact of the additional unexpected emissions to date has likely

**Table 1 Details of 3D model simulations.**

| | Model simulation | | | | | | | | | | | | |
|---|---|---|---|---|---|---|---|---|---|---|---|---|---|
| | CNTL | fODS | R2000_NoIE | R2000 | R2002 | R2009 | R2010 | R2000_CFC11_67 | R2000_CFC11_B | +R2000_CFC12_67 | R2000_CFC11_B | R2010_CFC11_B | R2000_NoVSLS |
| Purpose and description | Control | Fixed ODS | No increased emissions. CFC-11 emissions estimated by decay of 2002 bank only | Impact of meteorology. Simulations with repeating annual meteorology for years given below | | | | Impact of additional CFC-11 (and CFC-12) emissions. Simulations with CFC-11 scenario estimated from box model (CFC11_B) or from WMO (2018) at constant 67 Gg yr$^{-1}$ | | | | | Impact of VSLS. Simulation with zero chlorinated VSLS |
| Time period | 1980–2090 | 1960–2090 | 2000–2090 | 1955–2080 | 2018– | 2018– | 2018– | 2018– | 2018– | 2018– | 2018– | 2018– | 2018– |
| Meteorology | Varying | Varying | 2000 | 2000 | 2002/ 2003 | 2009/ 2010 | 2010/ 2011 | 2000 | 2000 | 2000 | 2000 | 2010/ 2011 | 2000 |
| CFC-11 emissions | WMO | 1960 vmr | Box model S_NoIE | WMO | WMO | WMO | WMO | 67 Gg yr$^{-1}$ | Box model | 67 Gg yr$^{-1}$ | Box model | Box model | WMO |
| Cl VSLS | Yes | – | – | – | – | – | – | – | – | – | – | – | No |

+ Also includes additional constant CFC-12 emissions of 59 Gg yr$^{-1}$ from 2018 onwards.

been very small and highlights the effectiveness of the atmospheric monitoring system for detecting small changes. If these continued emissions are related to production for an immediately emissive use (e.g. a solvent) then the overall impact on ozone may also be small. The potential impact from continued emissions is greater if they have arisen from much larger quantities of production for non-emissive use (e.g. foam). Run R2000_CFC11_B includes additional CFC-11 emissions based on the assumption that post-2010 use of CFC-11 was for non-emissive foam blowing (implying recent production that was around ×6 larger (see Methods) than the unexpected emission increase), and steady elimination of this production over the next 10 years. In this scenario, ozone loss is enhanced, causing a delay in the recovery of ozone of about 2 years no matter which metric is considered (Table 2). The simple WMO scenario of constant CFC-11 emissions of 67 Gg yr$^{-1}$ (S_CFC11_67) gives a different CFC-11 time dependence and much larger impact compared with the S_CFC11_B scenario. With large, constant emissions, the CFC-11 mixing ratio is maintained at higher levels late in this century (Fig. 2) and the ozone return to 1980 values (run R2000_CFC11_67) is delayed by around 18 years (Table 2, Supplementary Figure 6), in agreement with 2D model estimates with a less detailed treatment of polar processes[4]. However, we would note that this scenario of constant future CFC-11 emissions as used in WMO 2018[4] is likely unrealistic.

Chlorinated VSLS also affect polar ozone recovery through similar chemical processes. The impact of additional CFC-11 emissions can be compared with the results from run R2000_NoVSLS, which shows that if the stratospheric injection of chlorinated VSLS decreases to zero (from 2016 value of 114 pptv[30]) the ozone return dates are brought forward by about 7 years (Table 2), in broad agreement with the delay estimated by Hossaini et al.[18] who only considered $CH_2Cl_2$. This shows the leverage that chlorine from uncontrolled VSLS also exerts on polar ozone recovery. While the continued growth sensitivity scenario of Hossaini et al.[18] seems unlikely, further increases in short-lived chorine would delay ozone recovery in proportion to the increase in chlorine delivered to the stratosphere by these gases. Conversely, if the loading were to decrease from the present-day values ozone recovery would occur earlier. It should be noted that the impact on Antarctic ozone owing to changes in chlorine from CFC-11 (or CFC-12 or $CCl_4$; see the 'Discussion' section) or short-lived gases (e.g. $CH_2Cl_2$) is similar, and depends primarily on the amount of chlorine delivered to the stratosphere. The large mean age-of-air in the polar lower stratosphere (~5 years[31]) results in a large fractional conversion of most major organic chlorine (and bromine) source gases (whether long or short lived) to inorganic $Cl_y$ in air transported to this region[32].

The ozone-hole metrics in Fig. 3 can be transformed into a relative extent of ozone recovery by defining 0% recovery as the metric value at maximum depletion and 100% recovery as the 1980 value. This is demonstrated in Fig. 5 for run R2000, R2000_NoVSLS, R2000_CFC11_B and R2000_CFC11_67. This approach means that the extent of recovery at any time can be compared on a relative scale (see Table 3 for these values in 2050). Presentation of the results in this way avoids the issue of return dates being strongly affected by the shape of the ozone recovery trajectory, and that, under some circumstances, the atmosphere may not return to 1980 values at all. All simulations will give a numerical value for the extent of recovery at a given date and the scale of recovery is simply defined in terms of past ozone concentrations (e.g. 1980 levels and maximum depletion). The comparison date can be chosen as one that best suits the timescale of atmospheric processes and policy decisions. In 2050, the ozone mass deficit is 86% towards return to the 1980 value in R2000_NoVSLS, 76% in R2000 and 72% in R2000_CFC11_B but only 62% in run R2000_CFC11_67.

**Table 2 Dates for return to 1980 values for different ozone-hole metrics for 3D model simulations.**

| Ozone-hole metric | Model simulation[+] | | | | | | | |
|---|---|---|---|---|---|---|---|---|
| | R2000 | R2002 | R2009 | R2010 | R2000_CFC11_67 | R2000_CFC11_B | R2010_CFC11_B | R2000_NoVSLS |
| Minimum column ozone | 2063 | 2035 | 2062 | 2057 | 2078 | 2065 | 2058 | 2057 |
| Hole area | 2067 | 2041 | 2061 | 2056 | 2085 | 2069 | 2059 | 2061 |
| Mass deficit | 2067 | 2033 | 2061 | 2059 | 2083 | 2069 | 2060 | 2060 |
| September mean ozone column (90ºS–60ºS) | 2067 | 2021 | 2060 | 2052 | 2084 | 2069 | 2053 | 2060 |
| October mean ozone column (90ºS–60ºS)[*] | 2067 | 2031 | 2066 | 2085 | 2085 | 2069 | 2087 | 2060 |

[*]For comparison the multi-model mean results from Dhomse et al.[13] (MMM1S) for this metric are 2060 (2055–2066). + Simulation R2000_CFC12_67 includes additional constant CFC-12 emissions of 59 Gg yr$^{-1}$ from 2018 onwards. Recovery to 1980 values does not occur in this run by 2100, so the run is not included in the table.

**Fig. 4 Past and simulated future Arctic ozone showing the influence of meteorology and additional CFC-11 emissions.** Mean column ozone (DU) averaged from 90ºN to 60ºN for **a** February and **b** March from TOMCAT simulations CNTL (control), fODS (fixed ozone-depleting substances), R2000, R2002, R2009, R2010 (repeating meteorology from 2000, 2002/2003, 2009/2010, and 2010/2011, respectively), R2000_NoIE (no increased CFC-11 emissions), R2000_NoVSLS (no chlorinated very short-lived substances), R2000_CFC11_67 (with constant CFC-11 emissions of 67 Gg yr$^{-1}$), R2000_CFC11_B and R2010_CFC11_B (with additional CFC-11 emissions from box model for 2000 and 2010/2011 meteorology, respectively) (see legend) from 1960 to 2090. Panel **b** also shows mean (±1σ shading) chemistry-climate modelling initiative (CCMI) results from Dhomse et al.[13] and observations from NASA Solar Backscatter Ultraviolet (SBUV) instrument (black line). The coloured dots on the fODS line show the years used for simulations R2000, R2002, R2009 and R2010. The pink line in the background from 2018 to 2090 shows results of the continuation of run CNTL with 20-year repeating meteorology.

**Dependence on halogen loading and CFC-11 emissions**. The dependence of ozone recovery on the decline of halogens can help explain the difference of our Antarctic return dates with the CCM estimate of Dhomse et al.[13] of 2060 (2055–2066) and the impact of different CFC-11 and VSLS loadings (Table 1). Supplementary Fig. 3 shows temporal changes in both ozone and equivalent chlorine in the Antarctic diagnosed at 50 hPa in September and October. The post-2000 increase in ozone is driven by the decrease in equivalent chlorine. For equivalent chlorine, the only variation between the simulations is caused by differences in CFC-11 and VSLS, but ozone variability is also affected by meteorology. Supplementary Figure 4 compares Cl$_y$ and total chlorine from our CTM runs with CCM results[13]. For Cl$_y$ the individual CCMs show large variability and often underestimate observed values based on microwave limb sounder (MLS) data[33,34]. The Cl$_y$ values from the CTM runs (which also include

VSLS) are at the upper end of the CCM range and agree well with MLS. Apart from a few outliers, the total chlorine loading from the CCMs is less variable, indicating that although most CCMs have an overall realistic halogen loading for the past, the partitioning into Cl$_y$ in the Antarctic vortex is underestimated. In the future, the total chlorine (and Cl$_y$) loading in our CTM simulations is larger than the CCMs. This is partly due to the VSLS included in the CTM and partly due to the use of the updated WMO (2018) ODS scenarios (Supplementary Fig. 5), which delay the return of equivalent chlorine to 1980 values by about 4 years compared with the 2011 WMO Assessment[35] scenario used in CCMI.

Ozone return dates for the total column and at 50 hPa from the individual CCMs tend to correlate with Cl$_y$ return dates[13]. This is shown in Supplementary Fig. 7 along with our CTM results for varying meteorology and chlorine loading. Comparing the results

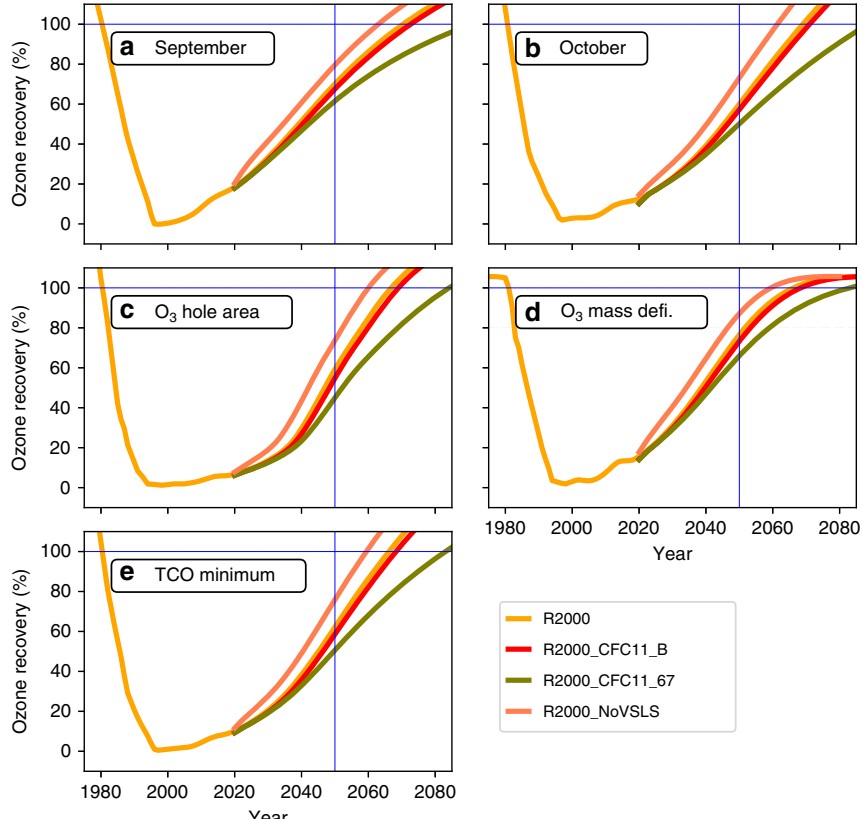

**Fig. 5 Extent of recovery for Antarctic ozone hole showing the influence of varying chlorine loading.** Extent of recovery (%) for the metrics of **a** September mean column ozone (90°S–60°S), **b** October mean column ozone, **c** area contained within the 220 DU contour (averaged September 7–October 13), **d** ozone mass deficit (averaged September 21–October 13) and **e** minimum column ozone (between September 21 and October 16) from TOMCAT simulations R2000 (2000 meteorology), R2000_CFC11_B (with additional CFC-11 emissions from box model), R2000_CFC11_67 (with constant CFC-11 emissions of 67 Gg yr$^{-1}$) and R2000_NoVSLS (no chlorinated very short-lived substances) (see legend) from 1980 to 2080. For the metrics 0% recovery is defined as the maximum depletion (which occurs around 1998) and 100% recovery is defined as return to the 1980 value. The blue horizontal and vertical lines indicate 100% recovery and 2050, respectively.

**Table 3 Percentage recovery (return to 1980 values) by 2050 from four model simulations with different chlorine loading for various ozone-hole metrics.**

| Ozone-hole metric | Model simulation | | | |
|---|---|---|---|---|
| | R2000 (%) | R2000_NoVSLS (%) | R2000_CFC11_B (%) | R2000_CFC11_67 (%) |
| Minimum column ozone | 61 | 75 | 57 | 49 |
| Hole area | 58 | 72 | 54 | 44 |
| Mass deficit | 76 | 86 | 72 | 65 |
| September mean ozone column (90°S–60°S) | 69 | 79 | 67 | 61 |
| October mean ozone column (90°S–60°S) | 60 | 72 | 56 | 49 |

from R2002 with R2010 shows the large impact of meteorology on the ozone return date. However, for a given meteorology, the variation of ozone return date appears to vary almost linearly with Cl$_y$ (e.g. R2000_NoVSLS, R2000 and R2000_CFC11_B) and to have a similar dependence (2000 vs. 2010 meteorology). Based on the different chlorine loadings in the simulations, the delay in ozone return date (Table 2) is around 10 years for an additional 150 pptv Cl. Thus, the impact of other CFC-11 scenarios on Antarctic ozone can be estimated without the expense of rerunning the full 3D model, as already noted for VSLS[18]. The simple dependence of ozone return on Cl$_y$ (whatever its source) is expected because of the role that chlorine plays in polar ozone loss cycles through the ClO + ClO and ClO + BrO catalytic loss cycles[36] and the small impact of climate change on Antarctic

ozone recovery[13]. The same dependency of ozone return on Cl$_y$ is likely to exist for the individual CCMs with realistic polar chemistry shown in Supplementary Figs. 4 and 7.

There is a compact, near-linear correlation between mean Antarctic column ozone depletion and the accumulated equivalent CFC-11 emissions (Fig. 6a, Supplementary Figs. 8 and 9), despite the different time evolutions of emissions in runs R2000_CFC11_B and R2000_CFC11_67, and the inclusion of CFC-12 emissions in run R2000_CFC12_67. This near-linear relationship can be explained by the behaviour of the ClO + ClO and ClO + BrO catalytic cycles for the relatively small perturbation of chlorine produced in the simulations[37,38] (see Supplementary Results 3). As expected, the runs with larger overall chlorine emissions give larger ozone depletion, but in the next few

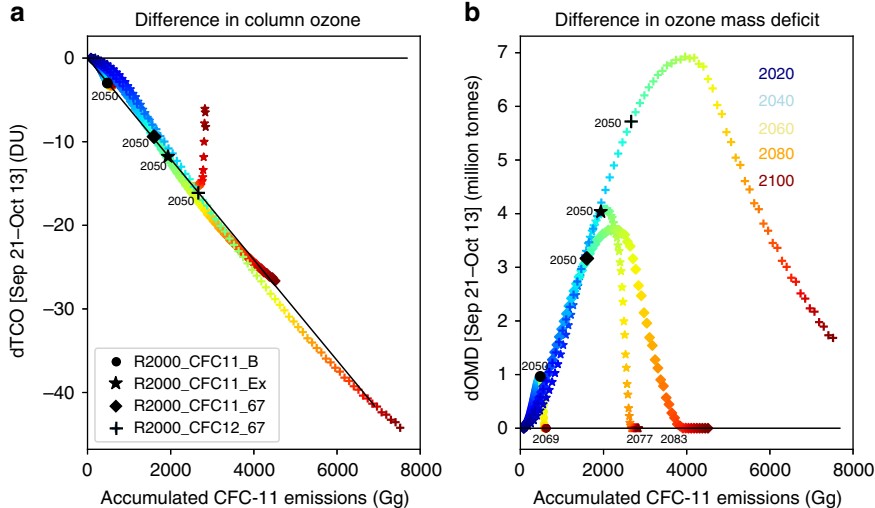

**Fig. 6 Antarctic ozone depletion versus accumulated chlorine emissions. a** The mean column ozone difference (DU) between run R2000 (2000 meteorology) and runs R2000_CFC11_B (with additional CFC-11 emissions from box model, circle), R2000_CFC11_67 (with constant CFC-11 emissions of 67 Gg yr$^{-1}$, diamond) and R2000_CFC12_67 (with constant CFC-11 emissions of 67 Gg yr$^{-1}$ and CFC-12 emissions of 59 Gg yr$^{-1}$, + symbol) in regions 60ºS–90ºS for the period September 21–October 13 (corresponding to the time period for the ozone mass deficit metric in Fig. 3) is plotted against accumulated additional equivalent CFC-11 emissions (Gg). Also shown are the results for simulation R2000_CFC11_Ex (star, see Supplementary Information). The colour shading indicates the year for each data point; the points for 2050 are plotted in black. The slope of best-fit line through the 2050 data points is 0.6 DU/100 Gg CFC-11. **b** Difference in estimated ozone mass deficit (million tons) versus accumulated equivalent CFC-11 emissions for the same simulations as panel (**a**).

decades the emission history does not play a role. For run R2000_CFC11_B, the emissions have decreased strongly by around 2080, and CFC-11 emitted early in the run has been removed from the atmosphere, hence the breakdown in the correlation on this longer timescale. The runs with fixed emissions also show a smaller deviation from linearity late in the century when some early emissions have been removed from the atmosphere. The slope of Fig. 6a (0.6 DU/100 Gg CFC-11) provides a means for estimating the Antarctic ozone impact of different emission scenarios. The additional CFC emissions increase the ozone mass deficit (Fig. 6b) and delay recovery. Despite the ongoing ozone depletion shown, runs R2000_CFC11_B and R2000_CFC11_67 are still estimated to have returned to 1980 values by 2069 and 2083, respectively (Table 2).

## Discussion

Recovery of the ozone layer is happening but any metric used to quantify its timescale or extent needs to be chosen with caution. For the Antarctic ozone hole, metrics related to the areal extent, ozone mass deficit or minimum column ozone appear more robust across different meteorological situations. In contrast, October mean column ozone can produce different year-to-year variations compared with other metrics due to meteorological variability in that month. This suggests a problem in using that metric to quantify ozone recovery from CCM simulations. Also, using return-to-1980-value dates can cause apparently large changes in recovery for small changes in column ozone[18]. A clearer picture of recovery under different scenarios can be obtained by estimating the degree of recovery achieved by a certain date. This can be designed to avoid over-emphasising the long tail in the recovery process when the ozone concentration is changing only slowly, but is very close to pre-ozone-hole values.

Whatever metric is used to estimate the recovery of the mean ozone layer, atmospheric variability will cause variations from year to year. It is very likely that the first year without an ozone hole (by the usual metrics) will occur well before the mean return date. For a

year with disturbed meteorology like that observed in 2002, then, by some metrics, the ozone hole may temporarily recover to 1980 conditions even with the enhanced halogen loadings of the 2020s. Clearly, an early year without an ozone hole (i.e. in the next decade or so) does not indicate full recovery from the effects of ODSs.

Renewed production and emission of CFC-11 will delay the recovery of the ozone layer. For the Antarctic ozone hole, there is a clear link between the additional amount of chlorine injected into the stratosphere, the additional polar ozone loss and the delay to recovery. However, even if the renewed production so far is for closed-cell foam use, immediate effective measures to stop this could imply a delay of just a few years. Should this renewed production be allowed to continue the impact will be correspondingly more severe, and for the Antarctic ozone hole, can be estimated directly from the chlorine-loading enhancement. The estimates derived here for the large WMO (2018) emission scenario of constant 67 Gg yr$^{-1}$ of a 18-year delay for an extra ~70-ppt CFC-11 are substantial but still do not change the overall trajectory of recovery. Note that for the Antarctic, the expected impact of climate change is only a small (2-year) advance of the 1980 return date[13]. In comparison, the potential impact of additional CFC-11 emissions can be large.

The relationship between Antarctic ozone recovery, emissions and polar Cl$_y$ loading can be used to apply our results to perturbations of other source gases. In the aged air of the polar lower stratosphere, most major source gases are nearly completely converted to inorganic forms. Therefore, if CFC-12 is being co-produced along with the additional CFC-11 and is largely being contained in a bank, a process that we have not explicitly considered, the impact on Antarctic ozone recovery will scale with chlorine loading supplied by this additional CFC-12 as it is eventually emitted. Minimum co-production rates of CFC-12 in the most common industrial process for producing CFC-11 are ~30%, so this additional chlorine might be substantial[39]. Similarly, the polar impact of any additional emissions of CCl$_4$, which is not decreasing as rapidly as expected in the atmosphere[40–42], will depend on the resulting increase in stratospheric chlorine.

Emissions of VSLS are also a source of stratospheric chlorine. Previous sensitivity studies[18] and our simulation R2000_NoVSLS demonstrate the potential for VSLS to influence timescales for ozone recovery through changing stratospheric chlorine loading. However, the degree to which such potential is realised and the overall significance of VSLS with respect to ozone will depend on both the magnitude and trend of their future emissions. For $CH_2Cl_2$, the most abundant chlorinated VSLS, global emissions increased by a factor of ~2 between 2000 (~500 Gg yr$^{-1}$) and 2016 (~1000 Gg yr$^{-1}$), as derived from NOAA data[4]. While such growth may have acted to offset the rate of upper stratospheric HCl decline, by ~15% since the mid-2000s[30], firm conclusions on any future impacts require more accurate estimates of likely future emission changes. For VSLS with both mixed emissive and non-emissive applications, notably $CH_2Cl_2$, such scenarios could be developed using production information and market analyses.

The Montreal Protocol is rightly seen as a seminal international agreement that has successfully led to decreasing levels of atmospheric chlorine and bromine and early signs of ozone recovery. For the Antarctic ozone hole, this recovery is best measured by metrics related to the extent of ozone loss in September rather than October and by the decrease in depletion by a certain date, rather than return to the 1980 value. Previous historic use of October metrics was related to datasets used to discover the ozone hole[2], but, with our detailed knowledge of the processes involved, we can now use different metrics to measure recovery (e.g. ref. [11]). In this paper, we have shown that three decades since it was ratified, its continued success does face some challenges from recent unreported CFC-11 production. However, with swift action to curb this production and any other, the long-term success of the protocol will be ensured.

## Methods

**Model configuration and experimental design**. We have used the TOMCAT/ SLIMCAT offline three-dimensional (3D) CTM to calculate the impact of chlorine scenarios and meteorological variability on stratospheric ozone[43]. The model has been widely used in previous studies[44,45] and simulates stratospheric ozone well. The model includes a detailed treatment of stratospheric chemistry, including a full description of processes related to polar ozone depletion. The model is forced by ERA-Interim reanalyses provided by the European Centre for Medium-Range Weather Forecasts (ECMWF)[46]. The model was integrated in series of experiments at a horizontal resolution of 2.8º × 2.8º with 32 levels from the surface to ~60 km. The runs were forced by observed and predicted surface mixing ratios of long-lived source gases from 1955 to 2100. The upper troposphere mixing ratios of short-lived chlorine species were specified using estimates from 2000 to 2016[30], with constant values before and after this period.

We drive the future model simulations with repeating meteorological analyses from previous years but with time-dependent ODS concentrations, which for the control run are taken from the baseline A1 scenario of WMO (2018)[4]. The surface $CH_4$ and $N_2O$ scenarios are taken from the Special Report on Emissions (SRES) scenario A1b (see Supplementary Information of Dhomse et al.[13]). This approach ignores the impact of climate change on stratospheric temperatures and circulation, which will become increasingly important as the simulations progress. Climate change certainly has an important impact on ozone recovery in the upper stratosphere where cooling acts to increase ozone and adds to the effect of decreasing chlorine[4]. However, in this paper, we focus on polar lower stratospheric ozone loss where the impact of temperature trends is less important[13]. Moreover, by using this approach, we ensure that the model results have realistic polar meteorological conditions that are important for accurate simulation of ozone loss. By running the model with different repeating analysis years we aim to span the effects of climate change on polar vortex dynamics.

Model simulation R2000 was integrated from 1955 to 2080 using repeating 2000 meteorology and time-dependent ODS concentrations (see Table 1). Control simulation CNTL was initialised from run R2000 in 1980 and integrated with varying meteorology until 2018. This simulation gives the most realistic representation of the atmosphere over the past 4 decades. Run CNTL was continued to 2080 using a cycle of 20 years of repeating meteorology from 1999 to 2018 (i.e. 1999 meteorology in model years 2019, 2039, 2059 and 2079). A series of future runs with repeating annual meteorology were initialised in 2018 from run R2000: R2002 (May 2002–April 2003), R2009 (May 2009–April 2010) and R2010 (May 2010–April 2011). These runs used repeating meteorology from May to April to avoid discontinuities in the polar winter/ spring of either hemisphere. Run R2000_NoIE is the same as R2000 but without the effect of the post-2002 stabilisation and post-2010 increase in CFC-11 emissions (box

model scenario S_NoIE in Fig. 1). Runs R2000_CFC11_B and R2010_CFC11_B were the same as R2000 and R2010, respectively, but with additional emissions of CFC-11 using box model scenario S_CFC11_B (Figs. 1 and 2). Run R2000_CFC11_67 was the same as R2000 but with constant CFC-11 emissions of 67 Gg yr$^{-1}$ (scenario S_CFC11_67, Figs. 1 and 2). Run R2000_CFC12_67 was the same as R2000_CFC11_67 but with constant CFC-12 emissions of 59 Gg yr$^{-1}$ (corresponding to equal numbers of molecules of CFC-11 and CFC-12 emitted). This is used as a sensitivity run to examine how the co-emission of longer-lived CFC-12 (lifetime 102 years[47]) will affect the model projections. Run R2000_NoVSLS was the same as R2000 but with post-2018 emissions of short-lived chlorine species set to zero (corresponding to 114 ppt less chlorine in the stratosphere). Finally, we performed a simulation fODS that was identical to R2000 but with halogenated ODS values constant at 1960 values. The length of the sensitivity runs varied depending on the rate of recovery to 1980 ozone values.

**Satellite data**. To compare with our past model simulations we use various satellite data products. For total column comparisons, we use the NASA Solar Backscatter Ultra-Violet SBUV (Version 8.6) merged dataset that is constructed by merging individual SBUV/SBUV/2 (total and profile ozone) satellite datasets[48]. SBUV-merged total ozone data are obtained from https://acd-ext.gsfc.nasa.gov/Data_services/ merged/. Values for Antarctic minimum ozone, ozone-hole area and ozone mass deficit averaged over defined time periods are obtained from https://ozonewatch.gsfc. nasa.gov/. For gridded total ozone data comparisons we use ESA's Copernicus Climate Change Service (C3S) data. These data combine ozone total column retrievals from various UV–nadir, limb and occultation satellite sensors. A key feature of these data is that climate data records (CDR) and interim-CDR parts of each product are generated using the same software and algorithms. Total ozone data used here are so-called level 4 data that combine data from 15 satellite instruments and fill the missing values using data assimilation system. These data are available from January 1970 to present at 1.0º × 1.0º resolution and are obtained from https://cds.climate.copernicus. eu/cdsapp#!/dataset/satellite-ozone?tab = overview

**Emissions box model**. We have related emissions of CFC-11 to mean atmospheric mixing ratios using a global 1-box model (Figs. 1 and 2). The model assumes a CFC-11 lifetime of 54.5 years, which was diagnosed from a TOMCAT full chemistry simulation. First, the model was used to relate the WMO (2018) global mean surface CFC-11 scenario[4] to an estimate of annual emissions, assuming that the surface values correspond to a global mean. This gave good agreement with independent estimates from a multi-box model[15]. Our estimated emissions were then used as a basis for estimating emissions from non-reported production of CFC-11 since 2010 and for future example sensitivity scenarios, bearing in mind the large uncertainties related to the new CFC-11 emissions and how they will vary. We use $13 \pm 5$ Gg yr$^{-1}$ as the estimate of emissions related to unreported production in recent years[15], which we relate to an assumed total production and project forward in time, assuming that the CFC-11 is produced for use in closed-cell foams. Scenario S_CFC11_B assumes that 15% of CFC-11 produced is released immediately[49] (ratio 5.66:1), followed by 3.5% yr$^{-1}$ (ref. [15]) as leakage from the bank. We assume that through policy action the unreported production will rampdown to zero over 10 years. Further box model runs were performed to examine the sensitivity to these assumptions (see panels in Figs. 1 and 2). The estimate of recent emissions was varied from 8 to 18 Gg yr$^{-1}$. Estimates of the ratio of production to initial emission are as large as 26 (ref. [49]), which seems inconsistent with likely applications. Therefore, we adopt the representative range of ×3 to ×9. The fractional release was varied from 2.5 to 4.5% yr$^{-1}$, which spans the upper range given in ref. [49]. Finally we varied the time for the production ramp-down from 5 to 20 years. A further future scenario was defined by taking the WMO (2018) scenario with an additional 67 Gg yr$^{-1}$ of emissions (the average estimated top–down emissions over 2002–2016, S_CFC11_67). This is likely unrealistic but it serves as a 3D model sensitivity simulation. Scenario S_CFC11_B leads to around 45-pptv additional chlorine in the atmosphere over the next few decades (Supplementary Fig. 5). The larger emissions of scenario S_CFC11_67 cause the CFC-11 decay to slow and by 2080 its vmr is around 170 ppt (510 ppt chlorine). A total of 4 CFC-11 scenarios used in the 3D CTM are shown in Fig. 2a, with one further described in Supplementary Results 1 and used in Fig. 6.

## Data availability

The observational datasets are available from the web links described in the text, i.e. SBUV-merged total ozone data from https://acd-ext.gsfc.nasa.gov/Data_services/merged/. Values for Antarctic minimum ozone, ozone-hole area and ozone mass deficit averaged over defined time periods from https://ozonewatch.gsfc.nasa.gov/; gridded total ozone data from ESA's Copernicus Climate Change Service (C3S) from https://cds.climate. copernicus.eu/cdsapp#!/dataset/satellite-ozone?tab = overview. The TOMCAT model results are available by emailing the corresponding author and via the web page http:// homepages.see.leeds.ac.uk/~lecmc/ftp/CFC11/.

## Code availability

The TOMCAT model is a research tool that is available to NERC-funded researchers in the United Kingdom and other collaborators who have access to suitable computing

facilities. The code is not otherwise publicly available. Reasonable requests to use the model should be made to the corresponding author.

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

## Acknowledgements

The TOMCAT modelling work was supported by the UK Natural Environment Research Council (NERC) through the SISLAC project (NE/R001782/1) and performed on the Archer HPC machine. We thank ECMWF for providing the ERA-Interim reanalyses. We acknowledge use of the publicly available C3S and SBUV data. RH is supported by a NERC Independent Research Fellowship (NE/N014375/1). J.K. and J.A.P. received funding from the European Community's Seventh Framework Programme (FP7/2007–2013) under Grant agreement no. 603557 (StratoClim).

## Author contributions

M.P.C. conceived the idea and initiated the study in discussion with S.D. M.P.C., S.D. and W.F. performed and analysed the model runs. The figures were prepared by S.D. and M.P.C. S.A.M. and J.S.D. provided guidance in scenario development. R.H., J.A.P. and J.K. provided comments on the model simulations. M.P.C. wrote the paper and included the comments from all of the coauthors.

## Competing interests

The authors declare no competing interests.
