## [Peer Review File · Nature Communications]

Reviewers' comments:

Reviewer #1 (Remarks to the Author):

Excellent, well written, paper discussing an important issue, namely the timing of the return of the stratospheric ozone hole to levels prior to human influence. The major contribution of this paper is to then examine how this recovery could be affected by further emissions associated with rogue use of chemicals like CFC-11 and the potential effect of short-lived gases not considered in the Montreal Protocol. This paper definitely should be published in Nature. However, I do think that the paper should add further discussion on uncertainties in their analyses, especially on the potential effects of climate change on their analyses.

Reviewer #2 (Remarks to the Author):

Review of the paper by Dhomse

This is a very interesting paper that should be published, but the paper would benefit from additional calculations to illustrate specific uncertainties and from substantial editing and modification.

My comments are as follows. Overarching comments are:

- 1) The paper presents results from a series of calculations. It was extremely difficult for me, even as someone who has followed the issues quite closely, to follow all the cases because there is nothing in the main text to succinctly summarize them; too much is relegated to the methods section and it isn't clear there either. The paper badly needs a descriptive table, to be included in the main body of the paper, presenting a compact description of the various cases to which the reader can refer.
- 2) There are many assumptions involved in the various cases. Many of them are presented as if they were subject to no uncertainties at all, when in fact they are highly uncertain. The paper's key figures and text would greatly benefit from giving ranges rather than separate lines or numbers reflecting the uncertainties in key factors. These include for example how much of the current mystery emission is extra production versus continuing emission from uncertain banks, how much of the implied extra production goes into foams rather than being released directly (again, not just a single number 1/7), etc. If the figures showed shaded lines instead of dotted and solid lines that could be one way to approach these in a clearer manner.
- 3) The paper is somewhat scattered in the way it goes from excess CFC-11 emission to uncontrolled VLSL. It may be better to put the latter in a later paper. If it stays here, more should be done to clarify what is new compared to their own group's earlier published work on the subject (by Hossaini et al.), and there may not be room to do so, particularly after other comments are addressed.

Specific comments are:

- 4) The paper contains some loose language such as that on line 21 'will largely cease to exist' (what does that mean?); in some cases accompanied by very specific numbers (2063-2068, same line). Please rework for clarity.
- 5) There are a number of unreferenced pieces of information. For example, I don't disagree that CFC-11 does represent about a quarter of the chlorine reaching the stratosphere (line 67) but where is the reference for this statement?
- 6) I understand lines 68-77 but I don't think the non-specialist reader will find the order of presentation clear. Move up lines 74-77 to the front of this part and then the rest can be made much clearer.

7) Line 91. What's new and different here?

8) line 111. Could it be delayed by more, or less, depending on the assumptions you have made? Stating this as if it were a single number when there are numerous assumption underlying your cases (see comment 2) seems misleading. Also on line 112, same point. Is this for the WMO scenario assuming all of the current emission is continued? That is somewhat extreme, isn't it?

9) Line 119-121. What's the point here? That the CTM produces a good lifetime so you get agreement? I don't know why this is relevant.

10) Line 126. There is uncertainty in the 2002 bank, and there is uncertainty in the release fraction from the banks, please include estimates of those.

11) Line 133-134. You've spanned some of them, but there are more. Please rephrase this to avoid giving the impression you've done a comprehensive study of uncertainty, unless your revised version does so.

12) Line 158. Some studies suggest significant depletion even before 1980. Do you mean first satellite data points? Then it could be given as 1979?

13) Line 172. This isn't a new point in the literature. Please refer to earlier papers.

14) Line 199. The comment about the free running models could be hard to understand for the non-specialist, and requires a reference as well. I think it's better deleted here.

15) Line 217-219. These are interesting results. It would be clearer as 'Whenever years with extremely cold stratospheric conditions occur in this century, the Arctic would be...'

16) Line 224-225. I think you mean 'and highlights the importance of how rapidly they were detected by atmospheric monitoring to safeguard the ozone layer'

16) Lines 225-227. Sentence is garbled.

17) Line 231. 7x is subject to large uncertainty as well as being unreferenced. Needs a range instead here.

18) Line 236. To say you have modeled the banks is a bit expansive. Better to say 'includes an estimate of the banks'. Hopefully one that includes uncertainties in some way in the next draft!

19) Line 239. The 2-D models are 2-D, but I don't think it's fair to say they have less detailed treatments of the chemistry. Rephrase.

20) Line 338-339. Confusing. Which case is this?

21) The figures are hard to decipher at present, in part because of the way different cases are labeled and in part because of the lack of a summary table describing the cases. For the key figure 1, I think it would be better to have three panels, one for each of the three scenarios. Even though they would be smaller they would be clearer. And they should have shading to show uncertainties, as discussed above. Could be split into two figures to avoid being too busy, emissions and CFC-11 mixing ratios, if one of the other figures is deleted (see below).

22) Figure 4 seems non-essential here; it could be moved to a supplement.

23) Figure 6 is interesting but a technical point that also could go to the supplement.

Reviewer #3 (Remarks to the Author):

Overall, this is a very well written paper which is rather easy to follow for a non-expert as I am. To my knowledge this is a solid piece of work, performed by a respected group.

My main points are:

Reading the paper I am wondering about the potential emissions of the unreported production and especially in relation to the bank that might have been formed. Assuming a 10 yr period (2010-2019) of unreported production leading to an average emission of 20Gg and 6/7 going in to the bank the bank would have grown by 120Gg per year to 1200. Am I correct that the loss rate of this bank has not been accounted for and may delay the recovery date further? Wouldn't this bank alone not already double the emissions for the coming decades (when I compare to 1480)?

Discussion: The sensitivity to the meteorological year is clear and will cause years with less and more strong ozone holes in the future with declining amplitude towards the recovery year. Given an unchanging climate this would give a random variability. The choice of indicator gives different horizons for the estimated recovery year but the systematic changes around the central estimates seem stable. Hence, the emission changes give rise to systematic changes in recovery estimate. Could you place these spreads in to perspective in comparison to the potential systematic impact of a changing climate? Would climate change and associated dynamics be significant compared to the unexpected emission impact? This issue could be handled in a few lines in the discussion. Now the upper stratosphere impact is shortly covered in the methods section.

Smaller remarks:

Line 19: changing chlorine emissions is obvious, so possible can be removed. Or do you mean not anticipated emissions deviating from the expected path?

Line 19/20: The term "chlorine source gas emissions" sounds a bit odd, would "active chlorine precursor emissions" or something alike be preferable?

Line 23: as 2021-2041 : doesn't this mean during any year from now on?

Line 56 Thus the averaging in this assessment of the interannual dynamical variability is not represented in the sigma range indicated above?

Line 66: here the accounted CFC-11 is meant

I apologize for the delay!

Response to Reviewers' comments, Dhomse et al.

We thank the reviewers for their helpful comments. These comments are reproduced below in *italics*, followed by our responses.

Reviewer #1

Excellent, well written, paper discussing an important issue, namely the timing of the return of the stratospheric ozone hole to levels prior to human influence. The major contribution of this paper is to then examine how this recovery could be affected by further emissions associated with rogue use of chemicals like CFC-11 and the potential effect of short-lived gases not considered in the Montreal Protocol. This paper definitely should be published in Nature. However, I do think that the paper should add further discussion on uncertainties in their analyses, especially on the potential effects of climate change on their analyses.

In this paper we chose to focus on polar (especially Antarctic) ozone depletion because it is dominated by halogen-catalysed loss and the impact of climate change on recovery is relatively small. Based on CCMVal-2 runs, Eyring et al (2010) did report a 1980 Antarctic springtime ozone return date about 10 years earlier due to the effect of climate change, which was ascribed to the impact of ozone increases in the upper stratosphere. However, the more recent study of Dhomse et al (2018) found a much smaller difference. We have added this to text. For additions related to other uncertainties please see responses to other reviewers below.

Reviewer #2

This is a very interesting paper that should be published, but the paper would benefit from additional calculations to illustrate specific uncertainties and from substantial editing and modification.

My comments are as follows. Overarching comments are:

1) The paper presents results from a series of calculations. It was extremely difficult for me, even as someone who has followed the issues quite closely, to follow all the cases because there is nothing in the main text to succinctly summarize them; too much is relegated to the methods section and it isn't clear there either. The paper badly needs a descriptive table, to be included in the main body of the paper, presenting a compact description of the various cases to which the reader can refer.

OK. Table 1 already listed results from most of the 3-D model runs. Therefore, it seemed most efficient to add extra rows and columns to this table to give details of all the 3-D simulations.

2) There are many assumptions involved in the various cases. Many of them are presented as if they were subject to no uncertainties at all, when in fact they are highly uncertain. The paper's key figures and text would greatly benefit from giving ranges rather than separate lines or numbers reflecting the uncertainties in key factors. These include for example how much of the current mystery emission is extra production versus continuing emission from uncertain banks, how much of the implied extra production goes into foams rather than being released directly (again, not just a single number 1/7),

etc. If the figures showed shaded lines instead of dotted and solid lines that could be one way to approach these in a clearer manner.

We agree that there are very many uncertainties in the various cases. Our aim is not to present a specific prediction of the impact of the renewed CFC-11 production on stratospheric ozone, even if we could present that with a large uncertainty range. Rather, we wanted to present some sensitivity estimates of the impact of these emissions on polar ozone. There are too many uncertainties in the production of CFC-11 – not least that the source is still unknown and the policy action to be taken has not been determined. Our aim is to frame the ozone impact in terms of chlorine loading, and (in the revised version) emissions. We have added text to emphasise the uncertainties and the aims of our study. Also, we have inserted a new Figure 6 which shows the impact on polar ozone in terms of accumulated emissions. In the discussion of this figure we consider the impact from a range of emissions, which could be very small (emissive use which stops soon) to much larger if significant production has gone into a bank. The presentation of our results in terms of CFC-11 emissions (new Figure 6) allows the assessment of other scenarios. Elsewhere in the responses we have addressed the specific issues of box model parameters being uncertain. The reason for using a representative simulation with treatment of a bank is to obtain a CFC-11 emission scenario which has a different time variation to the WMO constant emissions. We are then able to show the O₃ impact still varies near-linearly with the emissions (until the early emissions themselves are removed from the atmosphere). Similarly, we have now added a simulation with CFC-12, which has a longer atmospheric lifetime. Here the results in the polar region still remain near-linear.

3) The paper is somewhat scattered in the way it goes from excess CFC-11 emission to uncontrolled VLS. It may be better to put the latter in a later paper. If it stays here, more should be done to clarify what is new compared to their own group's earlier published work on the subject (by Hossaini et al.), and there may not be room to do so, particularly after other comments are addressed.

We have kept the VLS discussion in the text. We have used new observed estimates for the current loading of chlorinated VLS (Hossaini et al., 2019). Given the previous publications which have reported a large impact of VLS on ozone recovery, we think that it is useful to have these updated results in the same publication and with the same metrics as for CFC-11.

Specific comments are:

4) The paper contains some loose language such as that on line 21 'will largely cease to exist' (what does that mean?); in some cases accompanied by very specific numbers (2063-2068, same line). Please rework for clarity.

'largely cease to exist'. The disappearance of the Antarctic ozone hole will not be a simple switch from continual years with a hole to no further occurrences of a hole. Dynamical variability will mean that years with a hole will be interspersed with years without a hole. We tried to write something which conveyed the idea that by that time most years will be 'hole free'. To make this clearer we have added 'annual' and changed 'exist' to 'occur by'. We have changed the range to the approximate mid point (2065).

5) There are a number of unreferenced pieces of information. For example, I don't disagree that CFC-11 does represent about a quarter of the chlorine reaching the stratosphere (line 67) but where is the reference for this statement?

Sorry. The Montzka et al paper (referenced earlier in the paragraph) makes this statement in its abstract. Our sentence was a continuation of the discussion of that paper. We have now repeated the explicit reference to this paper and also clarified that this is a quarter of anthropogenic chlorine (about 20% of all chlorine – see WMO (2018), which is now also referenced).

6) I understand lines 68-77 but I don't think the non-specialist reader will find the order of presentation clear. Move up lines 74-77 to the front of this part and then the rest can be made much clearer.

We have considered the comment but decided to keep the original order, although with much editing to tidy up the text. We think that it is ok to first say that most CFC-11 produced ends up being emitted, and then mentioning the two possibilities. Because the new source is unknown we cannot assume that its use will follow what has happened before.

7) Line 91. What's new and different here?

The point we wanted to make is that Fang et al took the result of Hossaini et al but used them in a very simple way. They quoted a significant delay in return dates without showing a figure which illustrates how these large time differences occur (i.e. due to small shifts in lines with very shallow gradients). The point we wanted to make in these sentences is that it can be misleading to quote a large, headline delay to recovery to a fixed O₃ value, when you may, in percentage terms, be very close. That is an idea that we discuss further in our own results. To make the dependence of the Fang et al results to Hossaini et al we have changed 'cited' to 'used' and 'quoted' to 'thereby derived'.

8) line 111. Could it be delayed by more, or less, depending on the assumptions you have made? Stating this as if it were a single number when there are numerous assumption underlying your cases (see comment 2) seems misleading. Also on line 112, same point. Is this for the WMO scenario assuming all of the current emission is continued? That is somewhat extreme, isn't it?

Yes. This text has been deleted. It was included to fit into the Nature Communications style (of describing results at the end of the Introduction) but it is difficult to be concise and get across a useful message here. The main points and uncertainties are mentioned elsewhere.

9) Line 119-121. What's the point here? That the CTM produces a good lifetime so you get agreement? I don't know why this is relevant.

We need to explain how we derive the past emissions. We think it is useful to point out that our method (a simple box model) agrees with other estimates (e.g. multi-box models) in case someone questions the simpler approach. The phrasing of the sentence did, however, put the emphasis on the agreement. We have edited this so that the sentence states how we derive the past emissions. We have made the point about agreement with the multi-box model in the Methods.

10) Line 126. There is uncertainty in the 2002 bank, and there is uncertainty in the release fraction from the banks, please include estimates of those.

We have added Velders and Daniel (2014) has a reference for uncertainty in the Bank (cited in the Methods).

11) Line 133-134. You've spanned some of them, but there are more. Please rephrase this to avoid giving the impression you've done a comprehensive study of uncertainty, unless your revised version does so.

OK. We have deleted this sentence and added a few words to the previous one saying that the emissions increase may be smaller than we assumed.

12) Line 158. Some studies suggest significant depletion even before 1980. Do you mean first satellite data points? Then it could be given as 1979?

Yes, satellite data points as that is what is plotted. We have added the word 'shown'.

13) Line 172. This isn't a new point in the literature. Please refer to earlier papers.

We have added references to Revell et al. (2012) and Butler et al. (2016).

14) Line 199. The comment about the free running models could be hard to understand for the non-specialist, and requires a reference as well. I think it's better deleted here.

We have rewritten this as a full sentence and added the words 'assessment' and 'interannual' to the previous sentence.

15) Line 217-219. These are interesting results. It would be clearer as 'Whenever years with extremely cold stratospheric conditions occur in this century, the Arctic would be....'

OK. We have rephrased this.

16) Line 224-225. I think you mean 'and highlights the importance of how rapidly they were detected by atmospheric monitoring to safeguard the ozone layer'

No. We mean that the additional emissions have been detected at very small levels and before any measurable impact has occurred.

16) Lines 225-227. Sentence is garbled.

Rewritten with addition of word 'impact' and the sentence now starts 'The potential impact...'

17) Line 231. *7x is subject to large uncertainty as well as being unreferenced. Needs a range instead here.*

We have noted the large uncertainty in the $\times 7$ and added a reference to Ashford et al. (2004) in the Methods section.

18) Line 236. *To say you have modeled the banks is a bit expansive. Better to say 'includes an estimate of the banks'. Hopefully one that includes uncertainties in some way in the next draft!*

OK. We have edited the sentence.

19) Line 239. *The 2-D models are 2-D, but I don't think it's fair to say they have less detailed treatments of the chemistry. Rephrase.*

The point referred specifically to polar chemistry. Nevertheless the focus on chemistry is probably unfair as it will be in transport/dynamics were the 2-D formulation will have more difficulty in reproducing the 3-D atmosphere. Therefore, we have changed to 'polar processes'.

20) Line 338-339. *Confusing. Which case is this?*

This was based on the simulations shown in Figure S5 and the delays for run R2000_NoVSLs (7 years for 114 pptv Cl) and Run R2000_CFC11 (11 years for 50 pptv CFC-11 or 150 pptv Cl). We have added a sentence to explain this where Figure S5 is discussed.

21) *The figures are hard to decipher at present, in part because of the way different cases are labeled and in part because of the lack of a summary table describing the cases. For the key figure 1, I think it would be better to have three panels, one for each of the three scenarios. Even though they would be smaller they would be clearer. And they should have shading to show uncertainties, as discussed above. Could be split into two figures to avoid being too busy, emissions and CFC-11 mixing ratios, if one of the other figures is deleted (see below).*

We have added a summary table (see above) and improved the labelling of the cases. Given that, we have kept Figure 1 in the original style so that the panels are as large as possible and the cases can be intercompared.

22) *Figure 4 seems non-essential here; it could be moved to a supplement.*

As there is space in the main paper we have kept it there for now. We think it provides a useful visualisation of the magnitude of the impact.

23) *Figure 6 is interesting but a technical point that also could go to the supplement.*

This has been moved to the supplement (now **Figure SM5**). We have a new **Figure 6** which presents the polar ozone loss as a function of accumulated emissions, which helps to frame the discussion of the impact of uncertainties on our results.

Reviewer #3:

Overall, this is a very well written paper which is rather easy to follow for a non-expert as I am. To my knowledge this is a solid piece of work, performed by a respected group.

My main points are:

Reading the paper I am wondering about the potential emissions of the unreported production and especially in relation to the bank that might have been formed. Assuming a 10 yr period (2010-2019) of unreported production leading to an average emission of 20Gg and 6/7 going in to the bank the bank would have grown by 120Gg per year to 1200. Am I correct that the loss rate of this bank has not been accounted for and may delay the recovery date further? Wouldn't this bank alone not already double the emissions for the coming decades (when I compare to 1480)?

To clarify: We **do** account for the loss rate of the bank at 3.5%/year. Any renewed production which goes into the bank can be emitted at this annual rate. Therefore, the process described above has been accounted for.

Discussion: The sensitivity to the meteorological year is clear and will cause years with less and more strong ozone holes in the future with declining amplitude towards the recovery year. Given an unchanging climate this would give a random variability. The choice of indicator gives different horizons for the estimated recovery year but the systematic changes around the central estimates seem stable. Hence, the emission changes give rise to systematic changes in recovery estimate. Could you place these spreads into perspective in comparison to the potential systematic impact of a changing climate? Would climate change and associated dynamics be significant compared to the unexpected emission impact? This issue could be handled in a few lines in the discussion. Now the upper stratosphere impact is shortly covered in the methods section.

We chose Antarctic ozone as a case study because of the small impact of climate change on ozone recovery in this region. Ozone loss and recovery is dominated by chlorine and bromine loadings. We have added two sentences on this to the discussion. See also the response to Reviewer 1.

Smaller remarks:

Line 19: Changing chlorine emissions is obvious, so possible can be removed. Or do you mean not anticipated emissions deviating from the expected path?

We did indeed mean unexpected variations in emissions. We have added the word 'unexpected'.

Line 19/20: The term "chlorine source gas emissions" sounds a bit odd, would "active chlorine precursor emissions" or something alike be preferable?

We think that the term 'source gas' is common for stable gases which are emitted at the surface and reach the stratosphere. We have changed 'chlorine' to 'chlorinated' to make it clear that these source gases contain Cl and are not 'chlorine'.

Line 23: As 2021-2041: doesn't this mean during any year from now on?

This range represents the estimates from different metrics (see Table 1). The large range indicates how the different metrics can vary under particular meteorological conditions. In the case of 2002 meteorology, the September mean column ozone gives the earliest estimate of the return date. As the sentence relates to an ozone-hole-free-year, we have changed this to 'by around 2040 based on all usual metrics'.

Line 56 Thus the averaging in this assessment of the interannual dynamical variability is not represented in the sigma range indicated above?

Yes. We have added the words 'and is therefore not represented in the uncertainty ranges above'.

Line 66: Here the accounted CFC-11 is meant

Yes, in effect. The estimate is based on current lower atmospheric loadings, which overall have hardly been affected by the renewed production. This has been clarified.

Reviewers' comments:

Reviewer #1 (Remarks to the Author):

I found the authors to be quite responsive to the reviewer comments and they have revised the paper appropriately. I do not think further analyses or changes to the text are needed. I recommend going ahead with publication.

Reviewer #2 (Remarks to the Author):

Comments on the revised paper by Dhomse et al.

The revised paper has become clearer in several places, which is welcome. However, after a careful reading of the rebuttal and the revised paper I find that the authors did not address some key parts of my previous comments. I will be making them clearer here. Several major changes are necessary before this paper would meet the standards of Nature Communications. Most important, the test cases presented need major changes to avoid being misleading, which will propagate throughout.

My comments in order of their occurrence in the manuscript are as follows:

I will first discuss the title and abstract. The paper is not just about the unexpected emissions. It is about a series of factors that could potentially affect the recovery of the ozone hole (and a side statement about the Arctic) and how that recovery is estimated. Therefore, the subtitle to the title, "Impact of unexpected chlorine emissions" must be dropped to avoid being misleading.

Line 30. Simply adding although there are significant uncertainties' to the statement that 'disappearance of the ozone hole is ^{SEP}predicted to be delayed by ~10 years is an inadequate response to my earlier concerns. The figure of 10 years would undoubtedly attract significant attention if it is published in the abstract but it is not appropriate. There are very large uncertainties in past bank release fractions, current banks, and the amount of material assumed to be going into foams now that mean this number is likely much too large. The figure of 10 years should be removed, and replaced with a careful and well grounded estimate with an uncertainty range (i.e., not just a flat 10 years but language like $x \pm y$ when my comments below are included) that is at least plausible, with assumptions clearly stated. Of course it is not currently possible to fully quantify that range but that does not mean it is appropriate to give no uncertainties. Rather, it is all the more reason not to give a single value, to avoid implying that it's well characterized. I will comment further on this number and its associated assumptions below.

Line 31. If the impact is only linear over the next few decades, but the recovery is quite a bit later, how useful is this? If it's not an adequate guide for recovery then why is this statement in the abstract of a paper about recovery dates? Indicate the usefulness for this paper more clearly or remove.

Line 32-33 Regarding the short-lived chlorinated species, it remains unclear what the improvement is on the previously published work. Explicitly state what is new here and in the main text, or remove this to avoid being misleading about the context for this work.

Line 99. Needs a clear statement about what will be new and different here compared to this previous work, or remove entirely per the above.

Line 111. Makes it sound like the Arctic has recently often been extreme. Replace with a statement like 'based on the extreme meteorology observed in the year 2011.'

Line 117. Vague statement that does not communicate well. Replace "In this way our results can be scaled to other ^{SEP}possible emission scenarios." with a different statement that expresses what your new figure shows and is consistent with the statement in the revised abstract, if there is one (per my above comment).

Lines 127-159 Your statement that “The difference between the emissions from the bank and those derived from observations was used to estimate non-reported emissions from 2002 to 2017. This difference peaks at about 39 Gg yr⁻¹ in 2017” is remarkable. As presented, this would represent a massive amount of noncompliance to the Protocol that would be picked up widely and misused by scientists and press alike. Note that production and consumption are what is reported, not emission, so that language is not appropriate. While language can be fixed, that wouldn’t fix the problem. More important, a figure of 39 Gg of emission linked to additional unreported production is so important that it would require a detailed paper to justify it, which isn’t practical within the scope of the present paper. The uncertainties in current banks and their emissions (which are not “unreported”; they are allowed under the Protocol) are far larger than you represent them to be. A major revamp is therefore needed. All that is justified based on the available work (Rigby and Montzka) is their value for unexpected emission of the order 10 Gg/year (rather than 39 drawn from a box model), where their justification for saying that this represents unreported activity is the abrupt increase along with the change in the NH/SH ratio. Assuming that much more must be unreported is unjustified as well as misleading. One way to proceed would be to present calculations where you adopt the Montzka/Rigby quoted 13±5 Gg/year of emission believed to be associated with unreported production, i.e., only the recent increase and not the rest of the underlying curve of recent emissions. Then you can provide your figure of 7x more than that going into the bank, and you should also include some cases that represent uncertainties in that key number, such as 5x or 9x, and ending now or in 10 or 20 years, etc.). These cases should replace R2000_CFC11_B and R2010_CFC11_B throughout the paper and its supplement, and all associated discussion. This is a very substantial change but I don't see how it can be avoided if the work is to be convincing and appropriate given its scope and simple approach.

Line 187. In the Keeble study what was the measure? October monthly mean ozone hole size? Depth? Or?

Line 193-195. This important statement should be summarized clearly in the abstract.

Lines 263-266. Is this really needed? Why? What is new here compared to earlier work? State explicitly what is different here compared to, e.g., the 5 year figure given in Hossaini et al. or the 1 year and 5-9 year figures in Fang et al., and please quote those numbers to provide proper context for your work.

Line 292-341. This section seems out of place in the current paper. Although it is interesting, it is not very tightly tied to recovery dates, which is the subject of the paper. Clarify or remove. In addition, there are some technical points. Why Sep 21 to Oct 13? Would it be less linear if September were chosen, or October? Why? Also, why is this dependence linear when the reaction of ClO with itself is important, and might suggest a quadratic dependence? A lot of key things are left hanging here. Clarify or remove.

Line 361-362. Unclear what is meant here by “early year”. Clarify.

Lines 396-403: The discussion of the time to choose in identifying recovery is useful. However, the rationale for choosing October monthly mean column depth is partly historical and traces to the work of Farman et al., who only had data from a single station to use in identifying the onset of the ozone hole. The onset definition need not be the same as the recovery definition, however, and now we have much more information. This ought to be mentioned.

Figures.

I continue to think that the paper’s figures are not very easy to read. Figure 1 would be better if it were split up, and given the large uncertainties shading would be far better than lines.

Figure 4 does not seem very useful, and will be even less so when scenario R_2000_CFC11B is replaced with something more appropriate as outlined above. I think it should be removed or put in the supplement.

Response to Reviewers' comments (second round), Dhomse et al.

We thank the reviewers for their time and comments. The comments are reproduced below in *italics*, followed by our responses.

Reviewer #1

I found the authors to be quite responsive to the reviewer comments and they have revised the paper appropriately. I do not think further analyses or changes to the text are needed. I recommend going ahead with publication.

Reviewer #2

The revised paper has become clearer in several places, which is welcome. However, after a careful reading of the rebuttal and the revised paper I find that the authors did not address some key parts of my previous comments. I will be making them clearer here. Several major changes are necessary before this paper would meet the standards of Nature Communications. Most important, the test cases presented need major changes to avoid being misleading, which will propagate throughout.

>> We have redone the two major 3-D model test cases based on the box model scenarios and made related changes throughout. We have also addressed the reviewer's other comments. Please see responses below.

My comments in order of their occurrence in the manuscript are as follows:

I will first discuss the title and abstract. The paper is not just about the unexpected emissions. It is about a series of factors that could potentially affect the recovery of the ozone hole (and a side statement about the Arctic) and how that recovery is estimated. Therefore, the subtitle to the title, "Impact of unexpected chlorine emissions" must be dropped to avoid being misleading.

>> OK. We have deleted the subtitle as requested.

Line 30. Simply adding although there are significant uncertainties' to the statement that 'disappearance of the ozone hole is predicted to be delayed by ~10 years is an inadequate response to my earlier concerns. The figure of 10 years would undoubtedly attract significant attention if it is published in the abstract but it is not appropriate. There are very large uncertainties in past bank release fractions, current banks, and the amount of material assumed to be going into foams now that mean this number is likely much too large. The figure of 10 years should be removed, and replaced with a careful and well grounded estimate with an uncertainty range (i.e., not just a flat 10 years but language like $x \pm y$ when my comments below are included) that is at least plausible, with assumptions clearly stated. Of course it is not currently possible to fully quantify that range but that does not mean it is appropriate to give no uncertainties. Rather, it is all the more reason not to give a single value, to avoid implying that it's well characterized. I will comment further on this number and its associated assumptions below.

>> As noted below, we have changed two of the principal simulations in the paper based on smaller assumptions of additional CFC-11 emissions and discuss uncertainties using the box model (new Figures 1, 2 and S1). The results from these new simulations show a small delay in return dates of around 2 years. The simulation with the WMO (2018) constant CFC-11 emissions of 67 Gg/yr remains in the paper as it has been published in a prominent document and is therefore a reference within the community. We already say that based only on the additional emissions detected so far, the impact on ozone is negligible.

Based on this, we feel that it is not possible to give a number and range in the abstract. The lower limit is 0. In our model the WMO (2018) scenario causes a delay of around 18 years. Any range would have to encompass these numbers at least. We certainly take on board the reviewer's comment that we need to be careful not to give numbers which can be taken out of context. Therefore, we have edited the abstract to say (i) '*If these additional emissions cease immediately, the impact on stratospheric ozone will be negligible*', (having said that the additional emissions are the reported 13 Gg/yr), (ii) '*Assuming ... and cessation of this renewed production over 10 years, disappearance of the ozone hole is predicted to be delayed by a few years, although there are significant uncertainties*', and (iii) '*A scenario of constant, substantial emissions of 67 Gg/yr would delay Antarctic ozone recovery by well over a decade*.'

We think that this conveys the message that the impact of what is known so far, as well as summarising the impact of the published reference WMO scenario for which there are specific numbers.

Figures 1 and 2 now show sensitivity ranges for the CFC-11 box model scenarios. The impact of the assumed range on return dates is discussed for Figure S1 in the Supplementary Material. This range is small (± 1 year), compared to the large uncertainty in the timescale for rampdown if the WMO (2018) scenario of constant future emissions should occur, i.e. the problem is not addressed by policy makers.

Line 31. If the impact is only linear over the next few decades, but the recovery is quite a bit later, how useful is this? If it's not an adequate guide for recovery then why is this statement in the abstract of a paper about recovery dates? Indicate the usefulness for this paper more clearly or remove.

>> First, it is important to distinguish between recovery (a process which is underway) and the '1980 return date' which is an event which may occur decades from now. The above question combines the two notions. It is important to know how new CFC-11 emissions will affect stratospheric ozone, even if the effect does not persist long enough to affect a final return date. Additional ozone depletion in the near term will still have additional environmental impacts (e.g. increased surface UV). Policy decisions made now can affect how much additional ozone depletion will occur, therefore we believe it is useful to know how the impact on ozone varies with the CFC-11 emissions. We have modified this sentence to '*The impact on polar ozone over the next few decades scales with accumulated additional emissions, showing that any action to limit these renewed emissions will have positive environmental benefits*'.

Line 32-33 Regarding the short-lived chlorinated species, it remains unclear what the improvement is on the previously published work. Explicitly state what is new here and in the main text, or remove this to avoid being misleading about the context for this work.

>> OK. We have removed the comment on VSLs from the abstract, but kept mention of VSLs elsewhere. Chlorinated VSLs impact ozone in a similar way to CFC-11, i.e. by delivering Cl to the

stratosphere and we want compare this impact for a full range of ozone hole metrics and with up-to-date assessments of the VSLs abundances. The previously published work (Hossaini et al., 2017) only considered CH₂Cl₂, whereas here we have a more detailed treatment of many VSLs with estimates based on aircraft observations (Hossaini et al, 2019). Also, they only considered October return dates which gives a large impact for a small variation in ozone (as mentioned in the text) and can be misleading when quoted out of context (e.g. as in Fang et al., 2019).

Line 99. Needs a clear statement about what will be new and different here compared to this previous work, or remove entirely per the above.

>> Please see above response. Here we have added: ‘These results need to be reassessed for the impact of realistic amounts of chlorinated VSLs in the context of other changes to chlorine source gases and for a wider range of ozone hole recovery metrics.’

Line 111. Makes it sound like the Arctic has recently often been extreme. Replace with a statement like ‘based on the extreme meteorology observed in the year 2011.’

>> OK. We have added ‘observed in the year 2011’.

Line 117. Vague statement that does not communicate well. Replace “In this way our results can be scaled to other possible emission scenarios.” with a different statement that expresses what your new figure shows and is consistent with the statement in the revised abstract, if there is one (per my above comment).

>> We have changed the sentence to: ‘Our 3-D model results can therefore be scaled to assess the impact on column ozone of other total CFC-11 emissions’.

Lines 127-159 Your statement that “The difference between the emissions from the bank and those derived from observations was used to estimate non-reported emissions from 2002 to 2017. This difference peaks at about 39 Gg yr⁻¹ in 2017” is remarkable. As presented, this would represent a massive amount of noncompliance to the Protocol that would be picked up widely and misused by scientists and press alike. Note that production and consumption are what is reported, not emission, so that language is not appropriate. While language can be fixed, that wouldn’t fix the problem. More important, a figure of 39 Gg of emission linked to additional unreported production is so important that it would require a detailed paper to justify it, which isn’t practical within the scope of the present paper. The uncertainties in current banks and their emissions (which are not “unreported”; they are allowed under the Protocol) are far larger than you represent them to be. A major revamp is therefore needed. All that is justified based on the available work (Rigby and Montzka) is their value for unexpected emission of the order 10 Gg/year (rather than 39 drawn from a box model), where their justification for saying that this represents unreported activity is the abrupt increase along with the change in the NH/SH ratio. Assuming that much more must be unreported is unjustified as well as misleading.

One way to proceed would be to present calculations where you adopt the Montzka/Rigby quoted 13±5 Gg/year of emission believed to be associated with unreported production, i.e., only the recent increase and not the rest of the underlying curve of recent emissions. Then you can provide your

figure of 7x more than that going into the bank, and you should also include some cases that represent uncertainties in that key number, such as 5x or 9x, and ending now or in 10 or 20 years, etc.). These cases should replace R2000_CFC11_B and R2010_CFC11_B throughout the paper and its supplement, and all associated discussion. This is a very substantial change but I don't see how it can be avoided if the work is to be convincing and appropriate given its scope and simple approach.

>> OK. We have significantly modified the paper on this point. The 3-D simulations R2000_CFC11_B and R2010_CFC11_B have been replaced by new simulations with a different basis for deriving future CFC-11. The text in Methods has been fully revised. In summary, these new runs use the Monztkka/Rigby estimate of additional emissions associated with the unreported production of 13 Gg/yr. This is combined with estimates of the initial emission, emission ratio and fractional release (for references see Methods). Finally, we assume a 10-yr rampdown in production. Moreover, for these 4 parameters we now perform similar box model runs within a representative uncertainty range, and these ranges are presented as shading in Figures 1, 2 and S1.

The R2xx_CFC11_B runs show much less ozone depletion than the previous versions, in line with the much smaller emissions (e.g. see changes to Figures 3 and 5 and Tables 2, 3). The delay to the return dates is around 1-3 years (Table 1). In the revised paper, the WMO sensitivity scenario of constant 67 Gg/yr gives a much larger signal, and itself indicates a delay to return dates of around 18 years.

Line 187. In the Keeble study what was the measure? October monthly mean ozone hole size? Depth? Or?

>> The Keeble et al paper reported annual mean values. We have now clarified that and added in new analysis of their runs for October means.

Line 193-195. This important statement should be summarized clearly in the abstract.

>> It was covered in the abstract. Based on previous referee comments the relevant sentence in the abstract changed from:

“If the unusual meteorology of 2002 is repeated, an ozone-hole-free-year could occur as early as 2021-2041.”

in the first submission to:

“If the unusual meteorology of 2002 is repeated, an ozone-hole-free-year could occur by around 2040 based on all usual metrics.”

in the revised paper. The original range of dates was considered too wide. In view of the comment above we have added in mention of the possible early return by some metrics:

“If the unusual meteorology of 2002 is repeated, an ozone-hole-free-year could occur by around 2040 based on all usual metrics, and as soon as the early 2020s by some measures”.

Lines 263-266. Is this really needed? Why? What is new here compared to earlier work? State explicitly what is different here compared to, e.g., the 5 year figure given in Hossaini et al. or the 1 year and 5-9 year figures in Fang et al., and please quote those numbers to provide proper context for your work.

>> We have added a comment on being in agreement with the 5-year delay of Hossaini et al., bearing in mind the slightly larger Cl loading change of removing all VSLs, not just CH₂Cl₂.

Line 292-341. This section seems out of place in the current paper. Although it is interesting, it is not very tightly tied to recovery dates, which is the subject of the paper. Clarify or remove. In addition, there are some technical points. Why Sep 21 to Oct 13? Would it be less linear if September were chosen, or October? Why? Also, why is this dependence linear when the reaction of ClO with itself is important, and might suggest a quadratic dependence? A lot of key things are left hanging here. Clarify or remove.

>> The dates are the same as those often used for Ozone Mass Deficit (Figure 3, NASA website, WMO/UNEP assessments). We now state this in the caption of Figure 6 and we have put separate Sept / Oct plots in the Supplementary Material. Regarding ClO + ClO there are several points to note: (i) The additional Cl from CFC-11 is a small increment to the stratospheric Cly; the relative range in Cly is much smaller than for Cly from extra CFC-11. (ii) A large part of polar ozone loss is from the ClO + BrO cycles, which are not quadratic in [ClO]. (iii) It is known that ClO:Cl₂O₂ partitioning shifts as ClO_x (Cly) increases to favour Cl₂O₂. This damps out the square dependency (Searle et al., JGR, 1998). Overall this means there is no reason to expect a quadratic dependence. The 3-D model contains all of these factors and it is an interesting result that despite the range of factors, the impact scales in a compact, near-linear way. This is quite a technical discussion for the main paper and so we have added a single sentence there, and put a full summary of the above in the Supplementary Material.

Line 361-362. Unclear what is meant here by “early year”. Clarify.

>> OK. We have added ‘in the next decade or so’, which will certainly be a period when chlorine and bromine levels are still elevated.

Lines 396-403: The discussion of the time to choose in identifying recovery is useful. However, the rationale for choosing October monthly mean column depth is partly historical and traces to the work of Farman et al., who only had data from a single station to use in identifying the onset of the ozone hole. The onset definition need not be the same as the recovery definition, however, and now we have much more information. This ought to be mentioned.

>> OK. We have added two sentences in the final paragraph (lines 402-405).

Figures.

I continue to think that the paper’s figures are not very easy to read. Figure 1 would be better if it were split up, and given the large uncertainties shading would be far better than lines.

>> OK. We have split Figure 1 into two figures (for emissions and vmr, now Figures 1 and 2) and presented new sensitivity runs for 4 parameters in separate panels with shading. We have also greatly simplified each panel by removing most of the previous box model simulations. The main lines in the panels now relate to scenarios used in the 3-D model.

Figure 4 does not seem very useful, and will be even less so when scenario R_2000_CFC11B is replaced with something more appropriate as outlined above. I think it should be removed or put in the supplement.

>> OK. We have moved the figure to the Supplementary Material (now Figure S6). We have replaced the old R2000_CFC11_B with R2000_CFC11_67, as that shows a larger effect (and readers will be able to judge what a smaller impact from R2000_CFC11_B will look like).

REVIEWERS' COMMENTS:

Reviewer #2 (Remarks to the Author):

Review of the paper by Dhomse et al.

I appreciate the work done to respond to my comments. I believe that the revised paper is much more defensible as well as clearer. I have a few minor suggestions they may want to consider:

1) The title is improved but still does not fully correspond to the content of the paper, which is mostly about the Antarctic but also includes the Arctic (per its stated goal on line 113), and the contrast is useful. How about "How much longer will there be Antarctic and Arctic ozone depletion?"

2) lines 22-23 makes it sound like the 'some measures' are not among the usual ones but rather something exotic, which isn't so. How about "based on all usual metrics and as soon as the early 2020s by some of them".

3) line 33. Things can also scale in non-linear ways. How about 'scale linearly' to be clear.

4) line 54 would return rather than will return.

5) line 66-67. This paper later discusses partial recovery in some depth, so to avoid contradiction this would be better as "Indeed, until and unless ozone values return to the 1980 1980 values then recovery will be incomplete" or something similar.

Response to Reviewers' comments (third round), Dhomse et al.

We thank the reviewer for his/her time and final minor comments. The comments are reproduced below in *italics*, followed by our responses.

Reviewer #2

I appreciate the work done to respond to my comments. I believe that the revised paper is much more defensible as well as clearer. I have a few minor suggestions they may want to consider:

1) The title is improved but still does not fully correspond to the content of the paper, which is mostly about the Antarctic but also includes the Arctic (per its stated goal on line 113), and the contrast is useful. How about "How much longer will there be Antarctic and Arctic ozone depletion?"

>> We don't think that it helps to have 'ozone depletion' in the title. There will always be some potential ozone depletion from ODSs. In contrast whether there is a 'hole' or not depends on metrics which relate to e.g. 220 DU contour or 1980 values. Therefore, we expect a time when we can say that there is no longer an Antarctic ozone hole. We think that this notion will be easier for the Nat Comms readers to follow. As the paper is mostly about the Antarctic, and we have kept the term 'hole', we prefer to keep this in the title. Finally, we would note that we will also need to take on board comments from the Editor and consider what he thinks is most appropriate for the Nat Comms readers.

2) lines 22-23 makes it sound like the 'some measures' are not among the usual ones but rather something exotic, which isn't so. How about "based on all usual metrics and as soon as the early 2020s by some of them".

>> OK, text changed to 'some of them'.

3) line 33. Things can also scale in non-linear ways. How about 'scale linearly' to be clear.

>> OK, we have added 'linearly'.

4) line 54 would return rather than will return.

>> OK, changed.

5) line 66-67. This paper later discusses partial recovery in some depth, so to avoid contradiction this would be better as "Indeed, until and unless ozone values return to the 1980 values then recovery will be incomplete" or something similar.

>> OK. We have deleted 'at all'.